# Bridging the Gap: Using Wild African Elephant Metabolic, Reproductive, and Stress Hormone Profiles to Enhance Management in Human Care

**DOI:** 10.3390/ani15131863

**Published:** 2025-06-24

**Authors:** Kaitlyn M. Campbell, Chelsi J. Marolf, Peter E. Buss, Kari A. Morfeld

**Affiliations:** 1For Elephants, Inc., Omaha, NE 68134, USA; kcampbell26@huskers.unl.edu (K.M.C.); chelsimarolf@gmail.com (C.J.M.); 2School of Natural Resources, University of Nebraska, Lincoln, NE 68588, USA; 3Department of Biology, University of South Dakota, Vermillion, SD 57069, USA; 4Veterinary Wildlife Services, South African National Parks, Kruger National Park, Skukuza 1350, South Africa; peter.buss@sanparks.org; 5Morfeld Research & Conservation, LLC, Omaha, NE 68134, USA

**Keywords:** elephant, insulin, glucose, testosterone, progesterone, prolactin, leptin, cortisol

## Abstract

We aimed to establish baseline data for key metabolic, reproductive, and stress hormones in wild elephants, providing a critical reference for zoos to assess and refine their management strategies. Zoos have long faced challenges in managing the reproductive health and body condition of African elephants, with increasing concerns about the connection between obesity and infertility. Understanding these natural hormone patterns in wild elephants can help zoos adjust diets, exercise, and social management to better mimic natural conditions. This may improve overall health, body condition, and reproductive success for zoo-managed African elephants.

## 1. Introduction

Poor reproductive success in zoo-managed African elephants (*Loxodonta africana*) has been a recognized challenge for more than two decades [1]. Metabolic dysfunction, including obesity, has also emerged as a growing concern, and may be a major contributing factor to poor reproductive health [2]. Evidence for the effects of obesity and other metabolic perturbations in zoo elephants leads us to a critical question: What are the “normal” or “healthy” levels of these metabolic hormones in African elephants? A better understanding of the underlying connection between these metabolic and reproductive concerns in zoo populations requires studying their free-ranging counterparts. Several studies on wild elephants have aimed to identify the patterns of stress and reproduction [3,4,5,6], but no study has extensively examined metabolic hormones in free-ranging populations. Establishing hormone baselines in wild elephants will provide a valuable reference for evaluating whether the hormone patterns in zoo elephants differ from those in the wild.

It is well-recognized that obesity and metabolic conditions negatively impact the reproductive health of many species, including humans, companion animals, and domestic animals [7,8,9,10,11]. Zoo-managed species, including elephants, are often fed high-calorie diets and given inadequate exercise, which could result in metabolic perturbations [12,13,14,15]. Several studies have supported the concern that zoo elephants are obese, and that obesity is correlated with the high rates of reproductive complications. Morfeld et al. [16] found that the body condition scores (BCS) of non-cycling elephants were higher compared to cycling elephants. Investigators also revealed significant differences in the metabolic hormone concentrations between cycling and non-cycling females, with non-cycling elephants having higher concentrations of insulin and leptin and a lower glucose-to-insulin (G/I) ratio [2]. Similarly, Campbell et al. [17] found that a higher BCS and a lower G/I ratio were both predictive of lower testosterone concentrations in bulls.

These studies on zoo-housed elephants suggest that reproductive problems may be correlated with the metabolic imbalances associated with excessive body fat. This hypothesis is supported by research in horses and humans, which show that obesity can lead to metabolic changes that in turn impair fertility [18,19,20,21,22] In human males and females, glucose and insulin play critical roles in energy processing and metabolism [23]. Disruptions in the metabolic pathways can affect the hypothalamic–pituitary–gonadal (HPG) axis, which is partly responsible for regulating reproductive hormones, such as progesterone, estrogen, and testosterone [24]. For example, abnormal glucose or insulin levels can cause oxidative stress, which may reduce oocyte quality in females [25], and lower testosterone levels, impairing spermatogenesis in males [26]. Similarly, leptin, a hormone produced by adipose tissue, is also involved in regulating energy balance and reproductive function [27]. Elevated leptin levels, often associated with obesity, can lead to anovulation in female elephants [2], and may also impair sperm production in males, as has been observed in humans [28].

Stress hormones, such as cortisol, also have a complex relationship with metabolic health. Cortisol increases blood glucose levels by stimulating gluconeogenesis (glucose production in the liver) and reducing glucose uptake in peripheral tissues [29]. This process ensures energy availability during stressful events. However, chronic stress can lead to insulin resistance, where cells become less responsive to insulin—an early indicator of metabolic disorders in both humans [30] and other species [31,32]. Elevated cortisol levels can similarly reduce leptin sensitivity, increasing appetite and fat storage [33]. In domestic and zoo-managed animals, chronic cortisol production may contribute to metabolic imbalances similar to those observed in humans. Moreover, if stress is linked to metabolic hormone disruptions in managed elephants, it could help explain variations in weight gain and reproductive challenges in both sexes.

In the current study, our goal was to establish a wild elephant “baseline” for the G/I ratio, leptin, and their relationships with the reproductive hormones in females (progesterone and prolactin) and males (testosterone). We also included cortisol in our analyses. This will provide zoos with a much-needed reference to determine whether current mitigation strategies, such as diet and exercise aimed at restoring metabolic health and cyclicity, are effective or if alternative strategies are needed. Additionally, we sought to determine how metabolic, reproductive, and stress hormones (and their relationships) change with the season. Implementing management strategies that emulate the metabolic profiles of free-ranging elephants and consider natural hormonal variation due to seasonality will lead to a healthier, reproductively viable, and sustainable zoo population.

Using banked serum, we collected normative data from free-ranging African elephants (1) to determine concentrations of progesterone, prolactin, testosterone, cortisol, insulin, glucose, and leptin across various age classes and in both males and females, and to establish reference values for these parameters, (2) to assess the dynamics of these hormones due to seasonal effects, and (3) to examine the relationship between metabolic concentrations and reproductive hormones (progesterone, testosterone, prolactin) and stress hormones (cortisol).

## 2. Materials and Methods

### 2.1. Study Area and Population

Kruger National Park (KNP) encompasses approximately 19,485 km^2^ along the northeastern border of South Africa. Samples were predominantly collected from elephants in the southern part of the reserve, which is characterized by mixed woodlands with patches of thorn thickets and riverine forests. This region experiences distinct wet (October–March) and dry (April–September) seasons, with annual rainfall averaging around 500 mm. Average monthly rainfall data for the park was retrieved from https://www.sanparks.org/conservation/scientific-services/data-information-resources/kruger-climate-rainfall (accessed on 29 August 2024).

The total elephant population within the KNP is estimated to be more than 30,000 individuals [34,35]. Our study population included 87 female and 89 male African elephants, categorized as young (0–4 yo), sub-adults (5–15 yo), or adults (15+). The female cohort consisted of 28 young, 29 sub-adults, and 30 adults, while the male cohort included 30 young, 30 sub-adults, and 29 adults. Each elephant was represented by a single blood sample; no individuals were sampled more than once. Blood samples were opportunistically collected by South African National Parks Veterinary Wildlife Services between August 2000 and February 2016, and the serum was stored at −20 °C until analysis in 2019. The collection years for the serum samples ranged from 2000 to 2016, and for the adult males and females, the samples were limited to 2012 or later.

### 2.2. Hormone Analysis

The serum concentrations of androgen metabolites (testosterone), progestogen metabolites (referred to here as progesterone), prolactin, and glucocorticoid metabolites (cortisol) were analyzed using 96-well, multi-species enzyme immunoassays (K032-H1, K025-H1, K040-H1, K003-H1; Arbor Assays, Inc., Ann Arbor, MI, USA). Serum leptin was measured with an equine immunoassay (MBS1690910; MyBioSource, Inc., San Diego, CA, USA), and serum insulin was quantified using a bovine immunoassay (10-1113-01; Mercodia Inc., Uppsala, Sweden). Each sample was analyzed in duplicate, and absorbance readings were taken at 450 nm using a 4-parameter logistic curve-fitting software (Dynex Technologies Inc., Chantilly, VA, USA) and Revelation Quicklink software (version 4.2).

The serum progestogens, testosterone, cortisol, leptin, and prolactin assays were validated for use in African elephants by demonstrating parallelism between the standard curve and serial diluted serum samples. Significant recovery of the standards was also used to assess the accuracy of this hormone assay. Significant recovery was 99% for progestogens (y = 0.84x + 6.11), 99% for testosterone (y = 0.92x − 32.55), 99% for cortisol (y = 0.86x + 11.96), 98% for leptin (y = 0.82x − 0.26), and 98% for prolactin (y = 0.92x − 15.22). The insulin enzyme immunoassay had been previously validated for use in African elephants [2].

Serum glucose levels were measured using an automated glucose analyzer (One Touch Ultra, LifeScan, Inc., Milpitas, CA, USA) and glucose strips (UniStrip Technologies LLC, Charlotte, NC, USA, FDA 510(k):k113135). A glucose-to-insulin ratio (G/I) was calculated by dividing the glucose value (mg/dL) by its corresponding insulin value (μg/L; Morfeld, 2014 [2]).

### 2.3. Statistical Analysis

Data were analyzed in RStudio (version 2024.10.31) using R (version 4.4.2). Mean ± standard error was determined for each hormone for males and females, as well as subsets of each for adults, sub-adults, and the young. All data were standardized using the scale() function in R to eliminate effects of scale. Linear mixed models (LMMs) were performed in R with the lme4 package for each hormone and for glucose to determine potential relationships with elephant age class and with season (dry or rainy). Year was included in all models as a random effect, as we expect natural variation from year to year due to environmental factors. For instance, rainfall and therefore vegetation quality likely vary from year to year, and might impact metabolic profiles. We used the package emmeans to generate estimated marginal means.

As samples had a 16-year range in the collection date, we wanted to determine whether degradation had a significant effect on each hormone before any further analyses. Most hormones have not shown considerable degradation over long-term storage in previous studies [36,37,38], with many hormones stable for decades [39]. Serum metabolites are also stable in long-term storage at −80 °C [40], but we expect some degradation is possible in our samples due to storage at −20 °C [41].

Our adult male and female samples were collected only in 2012–2016, while the young and sub-adult samples were primarily collected prior to 2012. This resulted in collinearity between age and sample year as factors in our models. To assess degradation, we therefore first visually assessed each hormone and glucose in younger individuals (earlier sample years) and then in adults only. Only for glucose and insulin did we detect a pattern of lower levels during earlier years. We therefore assumed some degradation for glucose insulin. This may have resulted in an inflated G/I ratio in some older samples due to the scale of insulin. However, many of the samples in earlier sample years still had levels of glucose and insulin within a normal biological range. Additionally, there is little specific reference data in the previous literature on long-term storage effects for glucose and insulin. Without a consistent pattern of degradation, we therefore could not apply a blanket correction factor. This limits the interpretation in our results for glucose, insulin, and the G/I ratio.

Some extreme values were detected, and those outliers (values more than two standard deviations from the mean) were removed. Models were rerun when relevant and tables reflect the outlier-removed data. Our primary aim was to determine whether metabolic status could impact reproductive hormones. Because reproductive hormones should be highest in adults, we performed linear models for adults only for progestogens and testosterone to determine the potential effects of season, month, and rainfall on the concentration of insulin, glucose, G/I ratio, leptin, cortisol, progestogens, prolactin, and testosterone. For these multivariate generalized linear models (GLMs), the best model was determined using automated AICc model selection with the dredge() function in the MuMIn package [42]. Significance was set at α = 0.05 for all tests.

## 3. Results

### 3.1. Females

In the LMMs that included season and age class as predictors with sample years as a random effect, mean progesterone was significantly greater in adult females compared to young females (*t* = 3.240, *p* = 0.003) and sub-adults (*t* = 2.786, *p* = 0.011) when three outliers were removed (Table 1). In general, adults had more samples that were elevated. Lower (baseline) levels in these samples are expected to occur in females with normal ovarian cycles, which contribute to the large variation in our data (Figure 1a). Raw mean progesterone values were slightly lower during the rainy season (4.23 ± 3.93) compared with the dry season (6.36 ± 4.26), but this difference was not significant (*t* = −1.665, *p* = 0.202) and there was a considerable amount of variation. The cyclicity status of the females was not known.

One prolactin outlier (6.233 ng/mL) was removed from the analysis that may been from a pregnant adult female. Unpublished data from our lab using the prolactin EIA puts non-pregnant prolactin levels below 1–2 ng/mL. After removing this sample, prolactin did not vary with the season among the age classes but was significantly higher in adults than in the young (*t* = 2.418, *p* = 0.018) and sub-adults (*t* = 1.990, *p* = 0.050) (Figure 1b).

Cortisol was higher, though not significantly, in the young than in adult females (Table 1. Leptin was significantly lower in adults compared to the young (*t* = 3.220, *p* = 0.006) and sub-adults (*t* = 3.387, *p* = 0.006). The G/I ratio, however, was significantly lower in adult females compared to the young (*t* = 2.450, *p* = 0.017) and sub-adults (*t* = 2.819, *p* = 0.006), and it generally declined with age (Table 1, Figure 1c). However, as outlined above, we hypothesize that some of the variation in the female G/I ratio was due to degradation in the older samples, which were all samples from young and sub-adult elephants. Individual measures of glucose and insulin were not significantly different among the age classes, but generally insulin was higher in adults. Cortisol, leptin, and the G/I ratio did not vary with the season in females.

### 3.2. Males

In males, testosterone was significantly higher in adults than the young (Figure 2, Table 2; *t* = 3.700, *p* = 0.001 and sub-adults (*t* = 3.296, *p* = 0.004). Contrary to progestogens, mean testosterone was significantly elevated during the rainy season (16.2 ± 6.04) compared to the dry season (2.03 ± 0.33; Figure 3) (*t* = 2.895, *p* = 0.005). Cortisol, leptin, and the G/I ratio showed no significant variation among the age classes in males. The effect of the season varied by hormone, with no effect on leptin or the G/I ratio, but cortisol was significantly lower during the rainy season (*t* = −2.102, *p* = 0.0385) after correcting for age and sample year.

### 3.3. Relationships Between Reproductive and Metabolic Markers

In an adult-only model for males with testosterone as the response variable, only the G/I ratio remained as a significant predictor of testosterone after AICc model selection, though two competitive models also retained cortisol or season. However, the G/I ratio was a significant predictor of testosterone (*t* = 3.133, *p* = 0.004), explaining 27% of the variation in testosterone alone.

In the model for adult females with progesterone as the response variable, the best model included season and an interaction between cortisol and season. Competitive models also included the G/I ratio. During the dry season, cortisol had a significant negative effect on progesterone concentration (Figure 4, *t* = −3.374, *p* = 0.002), while only a slightly nonsignificant negative effect was observed for the wet season (*t* = −0.855, *p* = 0.401).

## 4. Discussion

The established ranges of the reproductive and metabolic hormones used for clinical reasons are heavily based on data from zoo-housed or semi-captive elephants. The data in this study therefore provide insight into the dynamics among the reproductive and metabolic health in natural populations of African elephants along with the potential impacts of the season. As expected, adult elephants in this study displayed higher reproductive hormones. The G/I ratio also varied with age. Many factors are related to reproductive and metabolic hormones in elephants, including environmental factors, such as season, rainfall, and vegetation [43,44,45], and social factors [46,47], as well as interactions among the hormones themselves [17,48,49].

### 4.1. Limitations

The current study utilized stored serum samples collected throughout Kruger National Park. Data collected with these samples did not include social parameters, behavioral characteristics (e.g., temporal secretions), or pregnancy status. We were therefore unable to determine the musth status for males or the cyclicity status for females, both of which would certainly impact hormones. Additionally, samples did not include the time of day.

### 4.2. Females

We used serum samples collected opportunistically over 15 years in 87 wild female African elephants to measure glucose, insulin, G/I ratio, leptin, prolactin, and progesterone. These data will provide useful baselines for zoo management and future studies on free-ranging elephants. We did not detect a strong effect of the season on any biomarker when taken across the age groups, but we did find differences with age. In adult females alone, we did not see a clear trend between the season and reproductive hormones, but we did find a relationship between increasing cortisol and decreasing progesterone during the dry season.

Metabolic markers are expected to vary with the season for wild elephants, as forage quality decreases and individuals switch from high quality grasses to lower quality browse from the wet to dry season, respectively [50]. Several studies have found glucocorticoid metabolites to be significantly higher during the dry season in African [4] and Asian elephants [6], which could be indicative of a stressful environment. Pokharel et al. [6] also observed that those with lower body condition scores had higher levels of a fecal glucocorticoid metabolite among free-ranging Asian elephants. Access to high-quality vegetation almost certainly affects body condition and metabolic stress. Wild, crop-raiding Asian elephants had lower glucocorticoids, which suggests that habitat quality might be even more important for glucocorticoid metabolite levels than the potential stress of human interaction [51].

In our data, insulin levels were moderately higher (with considerable overlap) and the G/I ratio was lowest in adult females compared to the other age groups in this study. Even with potential degradation in the young and sub-adult samples, this pattern is likely a true biological trend, though it may be slightly exaggerated in our data. In humans, insulin levels and glucose metabolism increase leading up to a peak around puberty [52], and then decline slowly throughout adulthood. Leptin similarly peaks at puberty and remains elevated in adulthood, and is thought to have a role in signaling preparedness for reproductive maturation [27,53]. Female African elephants reach reproductive onset in many cases before they are 14 years old [54,55]. The peak leptin levels observed in sub-adults in the present study therefore likely reflect reproductive maturity.

Leptin decreases after menopause in adult women, but in premenopausal women, leptin concentrations are up to 4x higher than in men [56]. In our sample population, leptin levels were lowest in adult female elephants, both compared to females of the other age groups and compared with adult males. All but four months of the rainfall data from the KNP were below average between April 2012 and December 2015, the period during which adult female serum samples were collected for this study. Lower leptin in adult females therefore could indicate poor foraging quality in these years due to drought and thus lower fat content in females. Regardless, increasing leptin levels in the current study were not negatively correlated with progesterone, as is commonly seen in zoo-housed females. The values reported here can therefore serve as a valuable reference.

Studies measuring the metabolic markers in zoo-housed elephants in the past decade have revealed correlations between obesity and reproductive acyclicity. Increased body condition scores, elevated leptin, and decreased G/I ratios are all correlated with a lack of a normal ovarian cycle [2,57,58]. Higher body condition scores, indicative of obesity, are likely in part due to a lack of exercise [16]. Zoos have already begun to address this issue [59], with many now achieving daily walking distance averages in zoo-managed elephants close to that of their wild counterparts [60,61]. A better understanding of the metabolic profiles throughout the lifespan of free-ranging African elephants may further help facilitate management practices by providing zoos with target metabolic profiles across all age groups.

Reproduction in females is costly, and given the distinct wet and dry seasons in South Africa, the timing of reproduction may be regulated by life history traits [62] and body condition. Indeed, studies have found glucocorticoid metabolites to be significantly higher during the dry season in African [4] and Asian elephants [6], which could indicate metabolic stress. Wittemyer et al. [43] also found that the season influenced the reproductive endocrine profiles of African elephants, with significantly lower fecal 5α-pregnane-3-ol-20-one during the dry season than the wet season, suggesting that ecological conditions, such as food availability, may impact reproductive hormones. We did not see a significant trend with rainfall or even between seasons in progesterone, but average progesterone was higher in the dry season, suggesting that seasonal dynamics may not always result in decreased progesterone. Wittemyer et al. [43] also found that the time since the last parturition negatively impacts 5α-pregnane-3-ol-20-one, which emphasizes the dynamic nature of progestogens and the many factors that could cause the variability seen in our samples.

Across all age classes, female African elephants in the current study did not display seasonal differences in reproductive or metabolic hormones. However, the negative correlation of cortisol on progesterone, particularly during the dry season, suggests that body condition and food resources may influence reproductive status, as suggested by others [3,43]. One of the biggest concerns with zoo-housed African elephant populations is the metabolic status of females, many of whom are overweight or obese [16]. In females, metabolic abnormalities such as obesity and elevated insulin and leptin are associated with acyclicity [2]. We did not find any significant relationships with leptin, insulin, glucose, or the G/I ratio and serum progestogens in our samples, but cortisol was a significant predictor during the dry season, which could reflect metabolic status in addition to the stress response.

### 4.3. Males

We analyzed serum samples from 89 free-ranging male African elephants to measure glucose, insulin, G/I ratio, leptin, cortisol, and testosterone. Our findings suggest that seasonality may influence gonadal and adrenal function in male elephants; however, metabolic hormone levels did not differ between the wet and dry seasons. And while metabolic biomarkers remained consistent across the age classes, the G/I ratio was a significant predictor of testosterone concentrations in adult bulls.

Across all age classes, testosterone levels were highest during the rainy season, while cortisol levels were lower. This contradicts a previous study that reported no clear seasonal pattern in fecal testosterone and glucocorticoid levels in free-ranging African bulls [63]. However, other research has linked the onset of the rainy season with increased musth occurrence—a reproductive phenomenon characterized by elevated androgen levels [45,64]. Supporting this, Ganswindt et al. [47] found that 61% of musth events occurred during the wet season and demonstrated that androgen levels were generally higher during this period. Although the musth status of the bulls in our study was unknown, this seasonal association could help explain the elevated testosterone concentrations observed during the wet season.

Glucocorticoid levels in bulls remained consistent across all ages; however, the average concentrations were higher during the wet season. In many species, the dry season is associated with increased physiological stress, likely driven by reduced food availability and distribution [65,66]. Presumably, male elephants in Kruger National Park may experience less nutritional stress, and therefore lower cortisol levels, during the resource abundant wet season. In zoo environments, most elephants do not experience seasonal fluctuations in food availability or quality, offering a unique opportunity to further examine how stress levels differ between managed and wild populations. Our findings could inform future studies aimed at understanding the interplay of seasonality, stress, and reproductive physiology across different captive settings.

Although both hormones varied seasonally, we did not observe a significant relationship between testosterone and cortisol concentrations. The association between these two hormones in male elephants has historically produced inconsistent results and is often linked to musth status. For instance, Ganswindt et al. [47] observed reduced glucocorticoid output during periods of heightened androgen production in free-ranging African elephants, whereas Ghosal et al. [5] found no difference in the glucocorticoid metabolites between musth and non-musth Asian bulls. In captive settings, Brown et al. [67] and Chave et al. [68] reported a positive correlation between the two biomarkers, while Campbell, et al. [17] found that cortisol did not significantly correlate with testosterone concentrations across the study population. These inconsistencies highlight the complex and context-dependent nature of reproductive and stress hormone interactions in male elephants.

Previous studies have similarly found mixed results for the correlation between testosterone and the G/I ratio. Our findings indicate that testosterone production increases with the G/I ratio in adult bulls. Campbell et al. [17] similarly identified a positive correlation between the two physiological indicators, but Chave et al. [68] found no significant relationship between the G/I ratio and testosterone when analyzing musth and non-musth bulls together, though they did observe a positive association when examining only non-musth bulls. This relationship is biologically plausible, as steroidogenesis, a metabolically demanding anabolic process, relies on sufficient energy availability. A higher G/I ratio generally reflects better metabolic health, as it suggests lower insulin levels and more stable blood glucose concentrations [69]. Conversely, a lower G/I ratio, associated with elevated insulin, may indicate insulin resistance and/or obesity, both of which are known to impair androgen synthesis across species [70,71,72].

This may also help explain our observation of higher mean G/I ratios in sub-adult and adult bulls compared to younger individuals, a trend notably opposite to that observed in females, and previously reported for managed bulls [68]. Musth, which generally becomes more pronounced with age, typically does not occur in its full form until around 25 years in free-ranging bulls [73]. Beyond the significant increase in testosterone, musth is also marked by reduced feeding and increased activity [73], often resulting in a decline in body condition [74]. In other species, similar fasting states lead to a gradual decrease in glucose and a more rapid decline in insulin [75,76], leading to an elevated G/I ratio. Under such energy-deficient conditions, a high G/I ratio may reflect a catabolic state rather than optimal metabolic health [77]. Musth is widely considered to be an energetically costly state [74,78], likely requiring a higher G/I ratio to sustain its duration.

Assuming musth contributed, at least in part, to the elevated testosterone production observed in adults, it is plausible that it also corresponded with the higher G/I ratio. Given that male elephants in human care have been observed exhibiting signs of musth as early as 13 years of age [79], it is important to recognize the role of metabolism in supporting reproductive health. Had musth status been included in our analysis, we may have been able to more clearly interpret these patterns. Nonetheless, our findings suggest that metabolic health is an important factor in maintaining the reproductive function in bulls, though further research is needed to clarify these complex relationships.

### 4.4. Comparison of Wild and Zoo Baselines

To better contextualize our findings and to evaluate their relevance to health outcomes in managed elephants, we compared glucose, insulin, and G/I ratios from our wild cohort with previously published values from zoo-managed populations [49]. Morfeld and Brown [57] used identical hormone analysis methodologies to report mean glucose, insulin, and G/I ratio values in 82 female and 23 male zoo-managed African elephants, ranging in age from 1 to 52 years. Unlike our study, however, their health assessments did not separate individuals by age class.

For zoo-managed females, the mean glucose concentration was 95 ± 21 mg/dL, the mean insulin was 0.65 ± 0.52 ng/mL, and the mean G/I ratio was 208 ± 113 [57]. By comparison, the combined averages for wild females across all age classes in our study were 85 ± 3 mg/dL for glucose, 0.58 ± 0.05 ng/mL for insulin, and 289 ± 38 for the G/I ratio. The 28% lower G/I ratio in zoo females could indicate insulin resistance, a potential early indicator of metabolic dysfunction [80].

For males, zoo elephants averaged 98 ± 19 mg/dL glucose, 0.56 ± 0.49 ng/mL insulin, with a G/I ratio of 326 ± 23, whereas wild males had slightly lower glucose (94 ± 3 mg/dL), insulin (0.47 ± 0.05 ng/mL), and a substantially higher G/I ratio (486 ± 61). These data suggest that wild elephants may maintain greater insulin sensitivity than their zoo-managed counterparts. Although differences in sample sizes, collection periods, and individual variability limit a direct comparison, the pattern is consistent with concerns about the impact of captive environments on metabolic health.

The apparent variation in the average G/I ratio between wild and managed elephants of both sexes warrants further investigation. African elephants generally reproduce successfully in the wild, and lower G/I ratios in managed individuals may contribute to the high prevalence of fertility issues observed in managed females and males under human care [81,82].

## 5. Conclusions

These findings underscore the importance of including free-ranging elephant data in shaping standards for elephants in human care. Wild elephants experience vastly different diets, exercise levels, and social structures compared to their managed counterparts, all of which may influence metabolic and reproductive function. Establishing hormone baselines provides a valuable reference for assessing the impact of zoo conditions on elephant physiology. This comparison is essential for identifying potential mismatches in metabolic and reproductive health, refining management strategies, and ultimately enhancing the well-being, fertility, and long-term sustainability of zoo-managed elephant populations.

## Figures and Tables

**Figure 1 animals-15-01863-f001:**
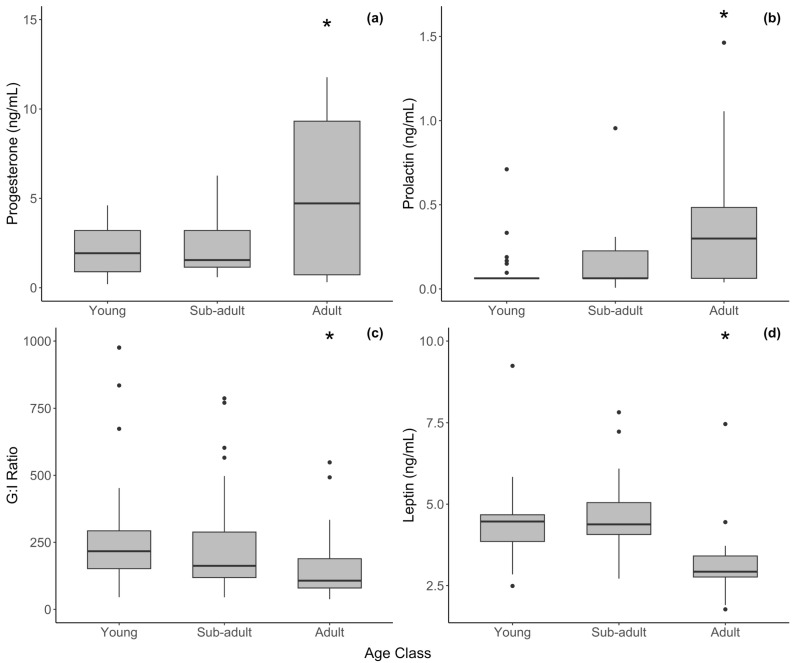
Hormone concentration by age class in female African elephants for serum (**a**) progesterone, (**b**) prolactin, (**c**) G/I ratio, and (**d**) leptin. The bold horizontal line represents the median, the shaded area represents the 25th and 75th interquartile range, and whiskers extend to the minimum and maximum values. Asterisks represent a significant difference between adult females and other age groups. Outliers were excluded from statistical analyses but have been retained in figures for transparency.

**Figure 2 animals-15-01863-f002:**
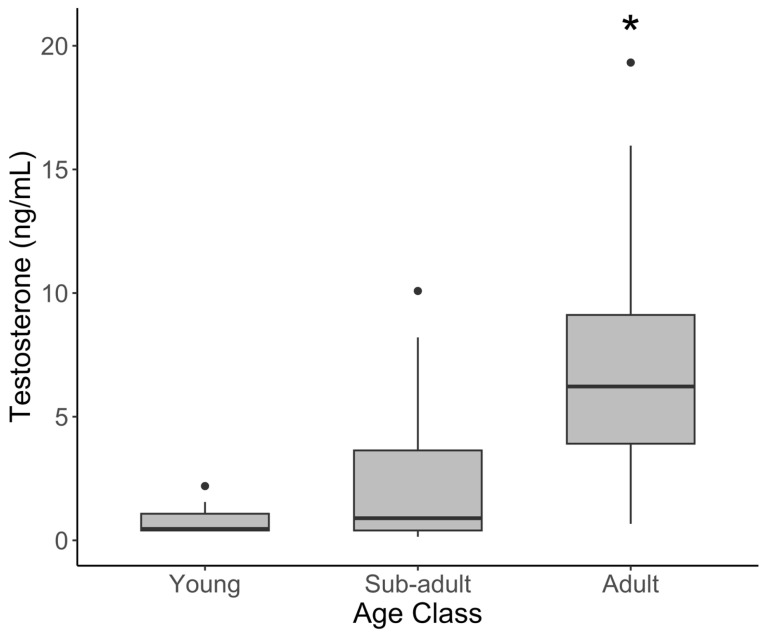
Serum testosterone concentration by age class in male African elephants. The bold horizontal line represents the median, the shaded area represents the 25th and 75th interquartile range, and whiskers extend to the minimum and maximum values. Asterisks represent a significant difference between adult females and other age groups. Outliers were excluded from statistical analyses but have been retained in figures for transparency.

**Figure 3 animals-15-01863-f003:**
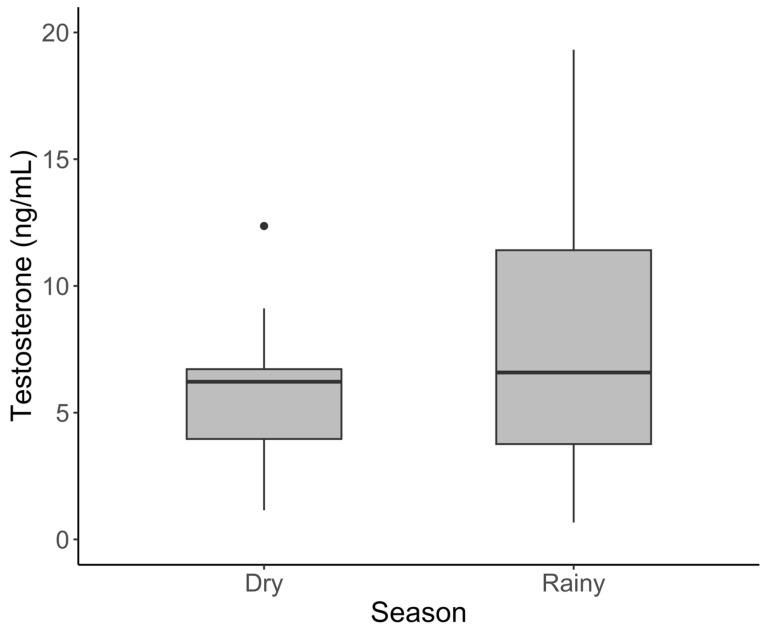
Serum testosterone concentration varied with the season in male African elephants. The bold horizontal line represents the median, the shaded area represents the 25th and 75th interquartile range, and whiskers extend to the minimum and maximum value. Outliers were excluded from statistical analyses but have been retained in figures for transparency.

**Figure 4 animals-15-01863-f004:**
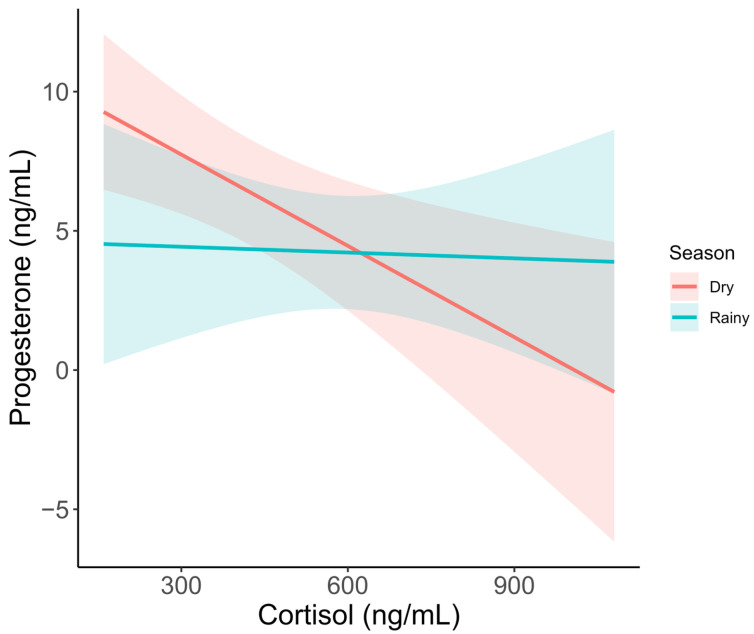
Serum progesterone metabolite concentration as it varies by cortisol and season. Trendlines show predicted values based on a GLM, including season and cortisol as predictors. Shaded areas represent 95% confidence intervals.

**Table 1 animals-15-01863-t001:** Summary table of minimum, maximum, and mean ± standard error for female hormone concentrations by age class.

Hormone	Age Class	N	Min.	Max.	Mean ± SE
Progesterone	Young	25	0.200	4.608	2.051 ± 0.251
(ng/mL)	Sub-adult	28	0.589	6.273	2.343 ± 0.302
	Adult	29	0.312	11.778	5.130 ± 0.752
Prolactin	Young	28	0.063	0.711	0.108 ± 0.025
(ng/mL)	Sub-adult	29	0.007	0.955	0.148 ± 0.033
	Adult	29	0.039	1.463	0.383 ± 0.066
Cortisol	Young	26	74.449	1293.850	703.151 ± 66.713
(ng/mL)	Sub-adult	28	181.655	1203.070	696.177 ± 49.513
	Adult	30	160.705	1075.590	503.823 ± 47.076
Leptin	Young	25	2.489	5.838	4.264 ± 0.155
(ng/mL)	Sub-adult	25	2.713	5.542	4.315 ± 0.135
	Adult	28	1.771	4.447	2.971 ± 0.113
Insulin	Young	28	0.038	1.534	0.430 ± 0.064
(mg/mL)	Sub-adult	29	0.050	0.987	0.388 ± 0.048
	Adult	30	0.050	2.780	0.904 ± 0.115
Glucose	Young	27	20.0	161.0	87.3 ± 7.0
(mg/dL)	Sub-adult	28	20.0	150.5	70.5 ± 5.5
	Adult	30	36.0	153.5	98.4 ± 4.4
G/I Ratio	Young	27	45.306	2342.105	355.813 ± 87.593
	Sub-adult	28	45.058	1160.000	317.577 ± 59.676
	Adult	30	37.950	1450.000	202.273 ± 48.506

**Table 2 animals-15-01863-t002:** Summary table of minimum, maximum, and mean ± standard error for male hormone concentrations by age class. NA refers to samples in which leptin was not measured.

Hormone	Age Class	N	Min.	Max.	Mean ± SE
Testosterone	Young	30	0.374	2.195	0.736 ± 0.083
(ng/mL)	Sub-adult	30	0.144	10.084	2.253 ± 0.476
	Adult	27	0.668	41.166	9.047 ± 1.708
Cortisol	Young	30	237.352	1496.071	750.182 ± 60.024
(ng/mL)	Sub-adult	29	167.340	1022.371	529.807 ± 44.888
	Adult	28	134.413	1703.508	714.923 ± 74.791
Leptin	Young	25	NA	4.064	NA
(ng/mL)	Sub-adult	NA	NA	NA	NA
	Adult	24	1.667	6.103	3.284 ± 0.231
Insulin	Young	30	0.050	1.238	0.372 ± 0.050
(mg/mL)	Sub-adult	30	0.044	1.715	0.388 ± 0.067
	Adult	29	0.031	3.000	0.697 ± 0.137
Glucose	Young	30	39.5	142.0	88.2 ± 5.2
(mg/dL)	Sub-adult	30	20.0	125.5	107.6 ± 3.4
	Adult	28	65.0	135.0	98.4 ± 4.4
G/I Ratio	Young	27	96.527	848.739	290.692 ± 33.664
	Sub-adult	28	34.985	924.000	417.223 ± 51.448
	Adult	23	38.667	591.324	221.935 ± 32.316

## Data Availability

Data are not available in an online repository but are available from the corresponding author upon reasonable request.

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
