# Peer review of "Bridging the Gap: Using Wild African Elephant Metabolic, Reproductive, and Stress Hormone Profiles to Enhance Management in Human Care"

_animals, 2025, doi:10.3390/ani15131863_

Round 1
Reviewer 1 Report
Comments and Suggestions for Authors
Overall this is a well written manuscript addressing important information needed about African elephants to facilitate information from wild populations to improve understanding and managed African elephant care. This reviewer recognizes the limitations of elephant research and commends the authors on utilizing banked serum samples in a manner that works to begin establishing important baseline data to allow this free population to contribute to the well being of managed populations.
Specific concerns listed in a line by line manner:
Line 44" "(Morfeld and Brown, 2014)" It appears that this should be cited as a reference. Is this reference 14 ? Please check all references.
Line 50: Please delete “in wild elephants” so this sentence flows more easily and is easier to read.
Line 69: In males and females of what species? The reference is human, not elephant. Please specify the species here.
Lines 75 -78: Leptin has been evaluated in elephants and is implied in the sentences as written, but the references for this section are general reviews or for humans. Consider adding the following references to this section and other appropriate places in the manuscript and rearranging references as necessary:
- Morfeld, K. A., & Brown, J. L. (2017). Metabolic health assessment of zoo elephants: Management factors predicting leptin levels and the glucose-to-insulin ratio and their associations with health parameters. PloS one, 12(11), e0188701.
*Note: this is reference 50 in the manuscript - Morfeld, K. A., & Brown, J. L. (2016). Ovarian acyclicity in zoo African elephants (Loxodonta africana) is associated with high body condition scores and elevated serum insulin and leptin. Reproduction, Fertility and Development, 28(5), 640-647.
*Note this is reference 36 in the manuscript
Line 119-124: 2 questions on methodology described here.
1) this is describing an analysis of samples nearly 20 years old. What is the evidence or references suggesting that there are not substantial alterations in the hormone levels measured as samples degrade over time when stored at -20C?
2) Why were samples limited for adults only? Was this because of differences in sample availability or were there changes in elephant populations that allowed more samples collected at the later years of the study period? The specifics of why there are differences in each group need to be expanded so readers understand why the adult samples are the most recent samples.
Line 156-157: Hormone degradation is mentioned as not being significant, yet how this level of degradation was determined was not mentioned. Please elaborate on this, as it is very important for this manuscript. How was hormone and glucose degredation determined?
Lines 163-164 There is a missing “were” between “Models” and “rerun”. 'Models were rerun when relevant and tables reflect outlier-removed data'
Line 169: GLM needs to be clearly defined somewhere. I do not see a clear definition of what a GLM is, but that acronym is used repeatedly.
Line 196: What does “was nearly significantly higher” mean? Something is either statistically significant or it is not statistically significant.
Line 202: Is age or year a predictor or are they a criteria that may affect hormone levels? What specifically is being predicted? I am unsure what the authors are predicting with an uncontrollable unknown variable in a study of banked serum samples. “Criteria” or “factor” seems to be more appropriate wording along with clearly defining what cant be determined.
Line 204 “generally declining with age” …. There appears to be a mixing of tenses in this sentence that makes it difficult to read. Adding “and it” to ‘generally declined with age’ creates a much easier to read sentence with consistent tense.
Section 3.2 Males (lines 208-215). I may be missing something in this description, but were the males grouped in the same way as the females? So was the same difficulty as described in lines 201-202 was encountered with males?
Lines 307 to 312… The authors provide details on leptin in humans with specific information about breeding age of the woman and how leptin levels change as they go from prepubescent to a peak at puberty to lower mature female to menopausal and cessation of natural reproductive ability. African elephants have been known to breed within the age span classified as “Subadult” in the manuscript---by 15 years of age. This beginning of breeding roughly corresponds with the time of puberty in humans, so it is expected that there would be lower leptin levels post puberty if elephants show hormone changes similar to humans. This also appears to correspond with data points described in Figure 1. This change in leptin levels should be discussed in more detail in the Discussion section of the manuscript. The information is in the manuscript, please bring it out a bit more so readers can clearly see that trend as female elephants mature.
Reviewer 2 Report
Comments and Suggestions for Authors
This manuscript describes the results of an important study that has implications for the management of elephants in human care. It is well-written and thorough, and I appreciate the authors attention to detail. I respectfully request that the authors consider the following minor comments/suggestions:
Materials and Methods: although the number of elephants studied in each category is listed later in a table, listing the total number of male and female elephants in this section would be helpful. In addition, I don't see where the number of samples analyzed for each elephant was listed. Including that information would strengthen this section of the manuscript.
Line 163: this appears to be an incomplete sentence.
Tables 1 and 2: asterisks are used to indicate that extreme outliers were removed, but they do not seem to refer to anything in the table. Were outliers removed in all measures in the tables? If so, then stating it as you have in the body of the manuscript seems sufficient.
Table 2: does "NA" mean "not applicable" or "not available"? Please explain why these data are not included.
Line: 389: does Campbell et al. make a claim of causation, or is the relationship they propose between cortisol and testosterone correlational?
Section 4.4: this section reads more like results to me, although I get the point you're trying to make. This seems like important information but does not come across as a particularly robust section of the manuscript. Perhaps some minor editing would more clearly highlight its importance.
